# Translation Control of *HAC1* by Regulation of Splicing in *Saccharomyces cerevisiae*

**DOI:** 10.3390/ijms20122860

**Published:** 2019-06-12

**Authors:** Xuhua Xia

**Affiliations:** Department of Biology, University of Ottawa, Marie-Curie Private, Ottawa, ON K1N 9A7, Canada; xxia@uottawa.ca; Tel.: +1-613-562-5718

**Keywords:** *HAC1*, *IRE1*, *ERN1*, XBP1, unfolded protein response, UPR, non-spliceosome splicing, ER stress

## Abstract

Hac1p is a key transcription factor regulating the unfolded protein response (UPR) induced by abnormal accumulation of unfolded/misfolded proteins in the endoplasmic reticulum (ER) in *Saccharomyces cerevisiae*. The accumulation of unfolded/misfolded proteins is sensed by protein Ire1p, which then undergoes trans-autophosphorylation and oligomerization into discrete foci on the ER membrane. *HAC1* pre-mRNA, which is exported to the cytoplasm but is blocked from translation by its intron sequence looping back to its 5’UTR to form base-pair interaction, is transported to the Ire1p foci to be spliced, guided by a cis-acting bipartite element at its 3’UTR (3’BE). Spliced *HAC1* mRNA can be efficiently translated. The resulting Hac1p enters the nucleus and activates, together with coactivators, a large number of genes encoding proteins such as protein chaperones to restore and maintain ER homeostasis and secretary protein quality control. This review details the translation regulation of Hac1p production, mediated by the nonconventional splicing, in the broad context of translation control and summarizes the evolution and diversification of the UPR signaling pathway among fungal, metazoan and plant lineages.

## 1. Introduction

Translation control occurs at different levels, with global and specific regulation as two extremes of the continuum. Global translation regulation impacts many genes. For example, dietary restriction in eukaryotes typically leads to a reduction of the ATP level and upregulation of translation repressor 4E-BP that binds to eukaryotic translation initiation factor 4E. This results in an overall reduction of cap-dependent translation. However, certain messages, e.g., translation of mitochondrion-targeted proteins, such as those in Complex I and IV of the electron transport chain, is increased [1]. In *Saccharomyces cerevisiae*, nutrient depletion also leads to a general reduction of cap-dependent translation, but a special set of genes that encourage invasive growth of yeast cells exhibits increased translation to take advantage of nutrients below the surface of the culture medium [2]. Another case of general translation control involves sulfur limitation, which reduces sulfur-containing amino acids such as methionine, which is needed for initiating translation [3]. Yeast cells respond to sulfur-limitation by using fewer proteins rich in sulfur-containing amino acid residues and replace the function of sulfur-rich proteins by sulfur-poor isozymes when possible [3]. 

Specific translation regulation involves a specific gene. Such specific regulation may be achieved through the action of trans-acting proteins [4] or microRNA [5], mediated by cis-acting nucleotide motifs. Autoregulation of several proteins represents prime examples of specific regulation. Eukaryotic genes encoding the poly(A)-binding protein (PABP) are known to have a poly(A) tract at 5’UTR, and PABP in high abundance will bind this poly(A) tract at 5’UTR to block translation by preventing the small ribosomal subunit from scanning to the initiation codon [6,7]. However, a poly(A) tract in 5’UTR that is shorter than the minimum binding size for PABP can recruit translation initiation factors and promote translation initiation [8]. This short negative feedback loop ensures optimal level of PABP production. Another example of translation autoregulation involves release factor 2 (RF2, encoded by *prfB* and decoding UGA and UAA stop codons) in many bacterial species [9,10]. The mRNA contains an inframe UGA stop codon followed by nucleotide C. UGAC represents a weak stop signal and is strongly avoided in highly expressed bacterial genes [11,12,13]. In *Escherichia cli*, when RF2 is abundant, this inframe UGA in *prfB* mRNA is decoded, leading to a truncated non-functional protein of only 25 amino acid residues. When RF2 is rare, this UGA is often not decoded, leading to a +1 frameshift by the ribosome and the production of a long functional RF2 of 366 residues. Thus, *prfB* mRNA is blocked from translation when RF2 is abundant, but allowed to be translated when RF2 is rare.

Translation regulation of *HAC1* mRNA in yeast (*Saccharomyces cerevisiae*) belongs to specific translation regulation. It is special because the regulation is achieved through splicing which occurs only during ER (endoplasmic reticulum) stress sensed by transmembrane kinase/endonuclease Ire1p. This review details the regulation of *HAC1* translation in the yeast, *Saccharomyces cerevisiae*, in the unfolded protein response (UPR) and outlines the diversity of the control mechanism in other evolutionary lineages.

## 2. *HAC1* and Its Translation Control via Ire1p-Mediated Splicing

*HAC1/YFL031W* [14] is located on chromosome VI of the yeast, and encodes a transcription factor of 238 amino acids that plays a key role in transmitting the signal of unfolded/misfolded protein response (UPR) from ER to the nucleus [15,16,17,18], regulating downstream UPR genes such as *KAR2, PDI1, EUG1, FKB2*, and *LHS1* that share a conserved UPRE motif [19]. Its pre-mRNA contains two exons and an intron of 252 nt (Figure 1). The intron has a peculiar 5’ss (G|CCGUGAU, where “|” marks the exon|intron boundary) instead of the consensus of G|GUAUGU in yeast. It ends with a CG dinucleotide (Figure 1) instead of the conventional AG dinucleotide. The intron is not processed by spliceosome, but is first cleaved by the protein kinase/endonuclease Ire1p [15,16], and then ligated by tRNA ligase Trl1p [20,21,22,23,24,25].

### 2.1. Intron-Mediated Translation Control of HAC1 mRNA

*HAC1* is constitutively transcribed [15,16,28], and its spliced and unspliced forms are termed *HAC1^i^* and *HAC1^u^*, respectively, with the superscripted “i” and ”u” for induced and uninduced [17,29]. Unfortunately, this terminology is not universally adopted, and the spliced and unspliced forms of *Xbp1* mRNA are represented as *Xbp1^s^* and *Xbp1^u^*, respectively, with the subscripted “s” and “u” for spliced and unspliced, respectively. *HAC1^u^* is exported to cytoplasm, contrary to most genes whose mRNAs are exported to the cytoplasm for translation only after splicing. 

While *HAC1^u^* and *HAC1*^i^ are equally stable [17], the presence of the intron in *HAC1^u^* inhibits its translation [16,17,30,31,32], whereas *HAC1^i^* is efficiently translated to generate functional Hac1^i^p. If *HAC1^u^* does get translated through experimental manipulation, e.g., by mutating the nucleotides involved in base-pairing in Figure 1 [33], the translation would stop at the stop codon highlighted in red in Figure 1. This results in a shorter protein (Hac1^u^p) missing the activation domain encoded in the second exon. Hac1^u^p exhibits much reduced transactivation activity on UPRE [28] than Hac1^i^p when assessed by using β-gal as a reporter gene [17,28,34].

The inhibitory effect of the *HAC1* intron on translation is hypothesized and experimentally verified [33,35] to be through long-range base-pairing between intron and 5’UTR [26,32,33], as schematically depicted in Figure 1. This base-pair interaction would interfere with the cap-dependent scanning by the small ribosomal subunit to find the start codon. This interpretation of the base-pair interaction depicted in Figure 1, suggesting inhibition at translation initiation, which is consistent with recent experimental evidence [33,35,36], but not with earlier hypotheses of inhibition at translation elongation [17,32]. These early studies claimed that *HAC1^u^* is associated with polysomes but its translation fails to complete because of ribosome stalling [17,32]. There are three lines of support for inhibition of translation initiation of *HAC1^u^* instead of ribosome stalling during translation elongation [35]. First, mutating sites involved in base-pair interactions increases *HAC1^u^* associated with polysomes [33]. Second, adding an inframe AUG at the 5’ end of the base-pair interaction releases translation block. Third, increasing RNA helicase eIF4A expression, which likely would facilitate removal of the base-pair interaction, reduces translation block. Three additional experimental studies [33,36,37] demonstrated that efficient ribosomal loading occurs only on *HAC1^i^*, but not *HAC1^u^*, consistent with the interpretation that translation block occurs during the initiation stage. In non-UPR cells where *HAC1* mRNA should almost all be in unspliced form, there are few *HAC1* mRNA in polysomes [33,38]. Di Santo et al. [33] provided direct experimental evidence to demonstrate that the polysome-like sedimentation of *HAC1^u^* is not due to ribosomes translating *HAC1^u^*, but is due to non-specific association between *HAC1^u^* and other actively translated mRNAs.

### 2.2. Ire1p-Mediated Splicing of HAC1 mRNA

As suggested in Figure 1, removing the *HAC1* intron would eliminate the translation block. The *HAC1* intron is not spliced by the regular spliceosome pathway, but is cleaved by the protein kinase/endonuclease Ire1p [15,16], first discovered in budding yeast cells experiencing accumulation of unfolded proteins [39,40]. The cleavage does not exhibit the regular order of step 1 reaction of cleaving the 5’ end of intron followed by the step 2 reaction of cleaving the 3’ end intron. Instead, the 5’ and 3’ ends are cleaved independently in random order [27]. The two exons are then ligated by tRNA ligase Trl1p [20,21,22,23,24,25]. This splicing not only removes the inhibitory effect of the intron on translation, but also replaces the last 10 amino acids (aa) encoded in *HAC1^u^* by the 18 aa encoded in the second exon. The last 18 aa function as a transcriptional activation domain [28] so HAC1^i^p features much higher transactivation activity than HAC1^u^p [17,28,34].

In response to accumulation of unfolded/misfolded proteins, the transmembrane Ire1p undergoes trans-autophosphorylation and congregates, in the presence of F-actin and a type-II myosin [41], into discrete foci of Ire1 oligomers within ER (Figure 2). The oligomerization is necessary for Ire1p-mediated splicing because the RNase activity of Ire1p is highly cooperative, so *HAC1^u^* cleavage is efficient only after Ire1p oligomerization [42,43]. It is at these Ire1 foci that *HAC1* pre-mRNA are spliced [26,42,43,44,45]. *HAC1* mRNA contains a 3’ cis-acting bipartite elements (3’BE, Figure 1) that signals to have *HAC1* mRNA carried to Ire1p foci for splicing. This 3’BE functions in a modular way. Inserting this 3’BE and a translation block into another RNA is sufficient to direct the recipient RNA into Ire1p foci [26]. 

The 3’BE serves not only as a signal for *HAC1* pre-mRNA to be carried to Ire1 foci for splicing, but also as a fail-safe mechanism to reduce accidental splicing of *HAC1* pre-mRNA causing unneeded UPR. When the cell is not in ER stress, *HAC1^u^* undergoes rapid 3’ to 5’ degradation to remove its 3’BE [46] so that the resulting *HAC1^u^* will no longer be carried to the Ire1p foci for intron removal (Figure 2). UPR-inducing agents decreases this 3’BE degradation [46] so *HAC1^u^* can be transported to Ire1p foci for splicing. This fail-safe mechanism should increase the reliability of the signaling pathway to avoid accidental splicing of *HAC1^u^* to trigger unneeded UPR.

### 2.3. Hac1p Triggers UPR and Is Autoregulated

Spliced *HAC1^i^*, generated in response to ER stress, is translated efficiently to Hac1^i^p [24,25,36,37]. Being a transcription factor, Hac1p contains a nucleus localization sequence (RKRAKTK located at sites 29-35, shared by both forms of Hac1p) [47] which directs Hac1p into the nucleus. Hac1p binds to unfolded protein response elements (UPRE, Figure 2) of its regulated target genes such as those encoding protein chaperones [48,49]. Because a number of UPR-related chaperone genes do not share the same UPRE, and may not be activated directly by Hac1p, and coactivators such as Gcn4p have been proposed [50,51]. One particular coactivator complex involved in Hac1p-mediated UPR is Gcn5p/Ada which increases histone acetylation at chromatin encoding UPR-related genes [24,25]. Negatively charged DNA is typically wrapped tightly around positively charged histone complex, hiding promotor sequences and transcription factor binding sites. Histone acetylation reduces the basic charge in lysine residues in histone so that DNA will not be tightly wrapped around histones to facilitate transcription.

Both Hac1^u^p and Hac1^i^p are rapidly degraded [17,34], with their half-life being about 1.5–2.0 min [17,34,47]. The short half-life of Hac1p indicates the importance or necessity of continuous production of Hac1p to sustain UPR. In this context, it is significant that the *HAC1* gene is autoregulated, harboring a UPRE at its own promoter region [52]. This generates positive feedback to produce Hac1p to sustain UPR, i.e., the more Hac1p, the more active transcription of *HAC1* and more Hac1p. This positive feedback is broken with dephosphorylation of Ire1p that reduces the Ire1p-mediated splicing of *HAC1^u^* [44,53] and rapid degradation of Hac1p.

## 3. Leaky Splicing of *HAC1* mRNA in Non-UPR Yeast Cells

Leaky splicing of *HAC1^u^* in non-ER-stressed cells is undesirable because it could lead to the production and accumulation of Hac1^i^p and accidental induction of UPR. Two mechanisms help to avoid leaky splicing. First, weak signaling by residual Hac1^i^p may be handled by Kar2p which is a chaperone protein. When Kar2p is free from the task of protein-folding, it binds to Ire1p to hinder Ire1p clustering and oligomerization so that the endonuclease activity of the RNase domain of Ire1p is not activated [54,55,56]. This works because the RNase activity of Ire1p is highly cooperative, so *HAC1^u^* cleavage is efficient only after Ire1p oligomerization [42,43]. Thus, a residual amount of Hac1^i^p would upregulated Kar2p production which would 1) reduce unfolded proteins, and 2) reduce Ire1-mediated splicing by hindering Ire1p clustering and oligomerization. Second, even if dimers or oligomers of Ire1p occasionally form, the 3’BE of *HAC1^u^* is efficiently removed by 3’ to 5’ degradation [46] to reduce the traffic of *HAC1^u^* towards Ire1p oligomers. 

In spite of the two fail-safe mechanisms mentioned above, leaky splicing of *HAC1^u^* may still occur. It is therefore interesting to characterize splicing efficiency of *HAC1^u^* in non-ER-stressed yeast cells. Empirical measurement of splicing efficiency (SE) of introns is based on the quantification of spliced (featuring exon-exon junctions) and unspliced (featuring exon-intron junctions) forms of individual genes or individual introns. An early attempt to characterize SE is by microarray, with exon-exon junction (EE) and exon-intron junctions at 5’ and 3’ sides of an intron (EI5 and EI3) as microarray probes to quantify spliced and unspliced form [57]. A simple but accurate method of quantifying SE by using RNA-seq data has been developed and applied to yeast introns [58]. The rationale is the same as the microarray approach. One obtains 1) the number of transcriptomic reads mapped to EE junctions (N_EE_) to quantify the spliced form, and 2) the number of reads mapped to EI5 and EI3 junctions of the intron (N_EI5_ and N_EI3_) to quantify the unspliced form. N_EI5_ is typically smaller than N_EI3_ because 1) step 1 splicing reaction occurs before step 2 splicing reaction so cleavage of EI5 junction occurs before EI3 junction, 2) enrichment of poly(A) mRNA by oligo-dT may lead to DNA replication reaching beyond EI3 but not EI5, and 3) 5’ degradation. Thus, the total mRNA (N_Total_) is measured not as N_Total_ = N_EE_ + (N_EI5_ + N_EI3_)/2, but as N_Total_ = N_EE_ + p × N_EI5_ + (1-p) × N_EI3_, with p estimated from data [58]. SE is defined as SE = N_EE_/N_Total_.

Most yeast genes, especially intron-containing ribosomal protein genes, are spliced efficiently (Table 1) with most mRNA in spliced form (large N_EE_ values) and less in unspliced form (small N_EI5_ and N_EI3_ values). However, *HAC1* mRNA is poorly spliced, with a small N_EE_ value of 32 and large N_EI5_ and N_EI3_ values (844 and 704, respectively, last row in Table 1). The result also confirms that 1) *HAC1* is transcribed constitutively in yeast [15,16], present in a significant amount in yeast cells not in UPR, and 2) *HAC1* splicing does occur in non-UPR cells so that rapid degradation of Hac1p [17,33,34,47] is necessary as a fail-safe mechanism against accidental triggering of UPR.

There is indeed weak splicing of *HAC1* in non-UPR yeast cells, with about 2% of *HAC1* in spliced form in non-UPR cells. UPR induction increases the percentage to ~69% in yeast cells. Thus, given the leaking splicing of *HAC1* mRNA, the few *HAC1* mRNAs observed in polysomes in non-UPR cells [33,38] might in fact be in spliced form.

The leaky splicing implies the production of Hac1^i^p and the risk of inducing unwanted UPR in non-stressed cells. Yeast has two fail-safe mechanisms against accidental induction of UPR. First, Kar2p, known to raise the threshold of UPR induction by participating in protein-folding and by hindering Ire1 from oligomerization, is directly and strongly upregulated by Hac1^i^p and would consequently suppress the weak signaling of UPR by a small amount of Hac1^i^p [54,55,56,60,61,62]. The second and perhaps far more important mechanism is the rapid degradation of Hac1^i^p. The degradation of Hac1^i^p occurs by the ubiquitin-proteasome pathway [33,47] and requires a nuclear localization sequence (29)RKRAKTK(35) encoded in the first exon [47]. If the degradation signal is encoded in the first exon, then it would explain why Hac1^u^p and Hac1^i^p are both highly unstable in the yeast [17,34], with the half-life being only about 1.5-2.0 min [17,34,47]. However, this turns out to be an incomplete story as is detailed in the next section.

## 4. Leaky Translation of *HAC1^u^*

Leaky translation of *HAC1^u^* was considered minimal given the well-documented intron-mediated translation inhibition (Figure 1), and the observation that Hac1^u^p is undetectable in non-ER-stressed yeast cells. This, together with the observation that Hac1^u^p is a much weaker transcription factor than Hac1^i^p when assessed by using β-gal as a reporter gene [17,28,34], does not suggest a risk of accidental UPR induced by Hac1^u^p. Thus, there seems to be no need for the budding yeast to evolve any protein-degradation mechanism specifically targeting Hac1^u^p. However, this reasoning turned out to be inadequate.

While leaky translation of *Hac1^u^* is indeed rare, there is substantial evidence that Hac1^u^p could potentially accumulate without efficient degradation [33]. Thus, rapid degradation is needed in non-stressed cells, not only for Hac1^i^p produced through leaky splicing and subsequently translation, but also for Hac1^u^p produced through leaky translation of *Hac1^u^*. In this context, a degron was discovered in the 10-aa encoded in the 5’ end of *HAC1^u^* intron [33]. In particular, this degron is recognized by Duh1p/Das1p, leading to a short half-life of Hac1^u^p of less than a minute [33].

It was previously thought that the degradation signal is encoded in the first exon, and therefore shared with both Hac1^i^p and Hac1^u^p, leading to both being equally highly unstable with half-life being 2 min or less [17,34,47]. However, if this were true, then there would be no need to have a degron in the last 10-aa of Hac1^u^p. It is therefore possible that the degradation signal in Hac1^i^p and Hac1^u^p are independently encoded, with a degron in the last 10-aa of Hac1^u^p [33] and a separate degron in the last 18-aa of Hac1^i^p. One may check if the degron in Hac1^u^p is conserved among different yeast lineages.

## 5. Ire1p Domain Structure and Its Splicing Activity

The luminal part of the transmembrane Ire1p can be roughly divided into two components [43,63,64]: the N-terminal domain with unsolved structure and the core luminal domain (N and cLD, respectively, in Figure 3). The N-terminal domain contains more positively charged residues than negatively charged ones (e.g., eight Arg and four Lys, but only three Asp and three Glu), leading to a high isoelectric point (pI) and positive charge under normal cytoplasmic pH range of 5.5–7.5 in yeast cells. However, the N-terminal domain also contains a consecutive stretch of amino acids (sites 8 to 30) that are highly hydrophobic (pointed by a red arrow in Figure 3). This hydrophobic stretch would need to be either buried deep into the folded protein or insert into a membrane. Given that the N-terminal appears structurally unresolvable [63], it does not seem likely that the hydrophobic stretch is buried deep inside a protein structure. Thus, the hydrophobic stretch might represent another transmembrane domain which would make Ire1p a double-pass instead of a single-pass transmembrane protein. However, this is a “backdoor” inference, i.e., one does not have direct evidence that the hydrophobic stretch is a transmembrane domain but makes the inference because an alternative is unlikely.

Multiple lines of empirical evidence supports the hypothesis that unfolded/misfolded proteins serve as direct binding ligands to the luminal domain of Ire1p to trigger Ire1p trans-autophosphorylation and oligomerization in both unicellular yeast [65] and multicellular eukaryotes [66]. The core luminal domain of the transmembrane Ire1p (cLD in Figure 3) is mainly responsible for oligomerization of Ire1p and formation of Ire1p foci to facilitate *HAC1* splicing [26,42,43,44,45] because Ire1p monomers have little splicing activity [42]. Note that, at a cytosolic pH of 7, a protein domain with a pI of 4.85 would be negatively charged, so cLD would tend to repulse each other against oligomerization. This may help avoid accidental oligomerization to prevent unwanted *HAC1* splicing and UPR, but would hinder Ire1p clustering at neutral pH. A cytosolic pH equal to 4.85 would reduce the cLD charge to 0, facilitating clustering and oligomerization. This suggests that UPR may be triggered more readily in an acidic environment. Acidosis does cause ER stress and induce UPR in human vascular endothelial cells [67]. *S. cerevisiae* can grow in a wide range of ambient pH, but the cytosolic pH is maintained in the range of about 5.5-7.5 [68,69]. Everything else being equal, one expects UPR to be induced more readily at low pH than at pH ≥ 7. 

One negative regulator of the IRE1p-HAC1p pathway is the luminal Kar2p [72] which binds to Ire1p under non-ER-stress conditions but dissociates from Ire1p under ER stress [60,61,62]. Kar2p is not essential for UPR induction but may serve to desensitize Ire1p to avoid unnecessary UPR [54,55,56]. The theoretical pI of Kar2p is only 4.62, and is therefore even more negatively charged than the cLD domain of Ire1p at neutral pH. This seems to create a challenge for Kar2p to bind to the cLD domain of Ire1p. However, while the cLD domain of Ire1 is overall negatively charged under neutral cytosolic pH, it has a stretch of positively charged amino acids, (91)RRANKKGRR(99), which may electrostatically interact favorably with the negatively charged Kar2p. Alternatively, the negative Kar2p could either bind to the positively charged N-terminal domain (Figure 3), or bind to cLM when the cellular cytoplasmic pH is near the pI of cLM and Kar2p so that they will not carry net negative charge. Given that *KAR2* is strongly upregulated by Hac1p [73] and that overexpression of Kar2p attenuates UPR [74], one may infer an alternative function of Kar2p as helping the cell exit UPR once ER homeostasis is restored. That the three regulatory motifs in *KAR2* that regulate *KAR2* expression independently [72] suggests additional functions of *KAR2*.

The cytosolic components of Ire1p include a linker domain that facilitates the docking of *HAC1^u^* to Ire1p, a kinase domain, and an RNase domain that cleaves the *HAC1^u^* intron [42]. The RNase activity is highly cooperative, so the cleavage of *HAC1^u^* is more efficient after oligomerization [42,43]. This series of Ire1p oligomerization, *HAC1^u^* docking and splicing should reduce the chance of accidental splicing of *HAC1^u^* by Ire1p [42].

The 30-aa transmembrane domain of Ire1p is highly hydrophobic (TM, Figure 3) and structurally unresolved as is typical of transmembrane protein domains. Its relatively high pI is mainly due to two residues that tend to be positively charged (one Arg and one Lys) and only one Glu that tends to be negatively charged. Its last 16 aa carry neither positive nor negative charges. 

The cytoplasmic linker domain (Figure 3) has a high theoretical pI (= 10.27) and is therefore strongly positively charged at neutral pH. This positive charge would facilitate the binding of the linker domain to negatively charged backbone of *HAC1* substrate. It is indeed crucial for the docking of *HAC1* pre-mRNA to the linker domain before cleavage by the RNase domain of Ire1p [42]. The linker domain also appears necessary for oligomerization in vitro [43].

The kinase domain of Ire1p features a stretch of negatively charged amino acids (624)DNDDADEDDE(633) almost immediately followed by a stretch of positively charged amino acids (641)KKKRKRGSRGGKKGRKSR(659). However, this segment of Ire1p is missing in the many structural studies on Ire1p. The kinase domain contains several phosphorylation sites (S840, S841, T844, and S850), so the empirical pI is expected to be lower than the theoretical pI of 7.24 (Figure 3). These phosphorylation sites serve as a genetic switch in the UPR circuit. When S840, S841 and T844 are mutated to Asp (which would be equivalent to permanent phosphorylation in terms of electric charge), yeast cells are unable to exit UPR after its induction [75].

## 6. How Is Ire1p Activity Regulated in Response to Unfolded/Misfolded Proteins in Yeast?

Many factors contribute to protein unfolding and misfolding. These include overexpression of an exogenous protein, e.g., a mouse protein in yeast cells [76], disruption of O-mannosylation in yeast [77], imbalance of ion concentrations such as calcium [78] and cadmium [79], excess of acetic acid, propionic acid and sorbic acid [80], antifungal agents that perturb protein folding such as monoterpene carvacrol [81,82], tunicamycin [18], dithiothreitol [83,84,85], hypoxia [86], or acidosis [67].

Ire1p senses ER stress caused by the accumulation of unfolded/misfolded proteins, and helps transmitting the signal of UPR from ER to nucleus by splicing of *HAC1* pre-mRNA (Figure 2). Ire1p activity of *HAC1* splicing can potentially be regulated through three mechanisms: (1) transcription, (2) translation, and (3) posttranslational modification. Although transcription and translation do not directly affect enzymatic activity of Ire1p, these two processes will increase Ire1p abundance on ER membrane and consequently higher probability of colliding with each other to oligomerize and activate the *HAC1-*splicing function. While transcriptomic characterization before and after UPR induction has recently been carried out for *S. cerevisiae* [73] and *Candida parapsilosis* [84], most studies focused on regulating Ire1p’s splicing activity by posttranslational modification, especially on autophosphorylation/dephosphorylation [87], which modulates *HAC1-*splicing activity of Ire1p and oligomerization, which is required for efficient Ire1p-mediated *HAC1-*spilcing [26,41,43,44,45].

There are multiple lines of evidence suggesting the *HAC1-*splicing activity of Ire1p is regulated by posttranslational modification. First, Ire1p-containing complexes do associate with RNAs in normal growing cells without UPR (UPR−), but these RNA species are longer than those found in Ire1p-containing complexes in cells under UPR (UPR+) [87]. This indicates (1) that splicing activity is absent in UPR− cells or at least weaker in UPR− Cells than in UPR+ cells and (2) that Ire1p-containing complexes have changed during the switch between UPR− and UPR+ states. Second, Ire1p has a cluster of phosphorylation sites: S840, S841, T844, S850, as well as a persistently negatively charged D836. Phosphorylation adds a negative charge to these sites and would potentially affect protein structure and its electrostatic interactions with its RNA substrates. Mutation at these sites in Ire1p, or replacing the negatively charged D836 by A836, reduces *HAC1* pre-mRNA splicing [88], although Ire1p clustering is not affected [42]. Third, protein phosphatase Dcr2p in yeast targets Ire1p specifically, physically interact with Ire1p’s phosphorylation sites S840 and S841, dephosphorylate Ire1p in vitro and downregulate UPR [53]. Another protein phosphatase Ptc2p also dephosphorylates Ire1p and negatively regulates UPR [44].

Exactly how cells exit UPR is not well elucidated, but three complementary mechanisms may operate for the purpose. The first is dephosphorylation of Ire1p [75] by either Ptc2p [44] or Dcr2p [53] or both. The three phosphorylation sites in the kinase domain (Figure 3), S840, S841, and T844, are particularly important for exiting UPR in yeast [75]. Phosphomimetic mutations of these sites to Asp results in failure for yeast cells to exit UPR after UPR induction [75]. The second is rapid degradation of Hac1p (Figure 2) which has a half-life of only about 1.5-2.0 min [17,34,47]. Hac1^u^p has a half-life even shorter because of the presence of a degron in its C-terminus [33]. The degradation of Hac1p is mediated by the ubiquitin-proteasome pathway [33,47] and requires a nuclear localization sequence (29)RKRAKTK(35) [47]. The third is the upregulation of Kar2p by Hac1p during UPR induction. Overexpressed Kar2p binds to Ire1p to attenuate UPR [74]. The time taken for yeast cells to exit UPR after its induction is in the order of hours [75].

## 7. Conservation and Diversification of Ire1p-Mediated UPR Signaling

There are three classes of potential cellular responses to the accumulation of unfolded/misfolded proteins: (1) reducing protein-folding demand in ER, (2) increasing protein-folding capacity in ER, and (3) apoptosis. Reducing protein-folding demand can be achieved by (1) selective mRNA degradation known as regulated Ire1p-dependent decay (RIDD) documented in both *Sch. pombe* [89,90] and mammalian cells [91,92], (2) reduced ribosomal proteins documented in yeast [93], (3) increased ER-associated degradation of unfolded/misfolded proteins known as ERAD [76,94,95], and (4) export unfolded/misfolded proteins to extracellular matrix [94]. Increasing protein-folding capacity in ER can be achieved by (1) the Ire1p-Hac1p signaling pathway to increase the production of chaperone proteins [48,49] or (2) direct processing and stabilization of mRNA encoding chaperone proteins, such as *BIP1* mRNA in *Schizosaccharomyces pombe* [89]. These responses are often referred to as adaptive because they all contribute to restoring ER homeostasis. When all these fail to restore ER homeostasis after prolonged UPR, apoptosis is triggered [79,85,96,97,98].

These cellular responses are not independent of each other. For example, RIDD is involved in (1) restoring ER homeostasis when it degrades mRNA encoding ER-translocating proteins to reduce protein-folding demand, and (2) apoptosis when RIDD eventually begins to degrade anti-apoptosis miRNAs after prolonged UPR failing to restore ER homeostasis. Two characteristic events occur during the transition of RIDD from contributing to ER homeostasis to apoptosis. First, it degrades *KAR2* mRNA and consequently increases ER stress [97]. Second, it cleaves anti-apoptosis microRNAs such as miR-17, miR-34a, miR-96 and miR-125b allowing apoptosis gene products to accumulate [97,99]. There might be a switch between adaptive response and apoptosis [79,85,96,97,98].

While Ire1p-mediated UPR contributes to all these responses, individual species may possess only a subset of responses. For example, RIDD is a dominant response in UPR to restore ER homeostasis in *Sch. pombe* which has Ire1p but no Hac1p homologue [89,90]. RIDD is also well developed in many multicellular species that do have Hac1p/Xbp1 homologues, including insects, mammals and plants [91,92,100]. However, RIDD is missing in *S. cerevisiae* as *Hac1* mRNA is the only detected substrate of Ire1p cleavage [100]. Thus, while many species feature both RIDD and Ire1p-Hac1p/Xbp1 pathways, *Sch. pombe* has lost the Ire1p-Hac1p/Xbp1 pathway, and *S. cerevisiae* has lost the RIDD pathway. 

### 7.1. Conservation and Diversification of the Ire1p-Hac1p Pathway

The key genes involved in UPR signaling in the yeast are *IRE1* and *HAC1*, together with genes such as *TRL1* that helps ligate cleaved *HAC1* exons into a processed and efficiently translatable mRNA (Figure 2). The evolutionary conservation of this UPR signaling pathway is best exemplified by *IRE1* which has homologues (e.g., *ERN1* in mammals) in many studied eukaryotes [101,102] including mammals [103] with universally conserved function of participating in the unconventional splicing of pre-mRNA of *HAC1* or its functional homologues such as Xbp1 [104,105] in metazoans and bZIP60 in plants [101,106]. 

The *IRE1+HAC1* combination in yeast and *Ire1+Xbp1* in metazoans work analogously. For example, in both *Caenorhabditis elegans* and mice, *Xbp1* splicing occurs during UPR, with proteins from spliced *Xbp1* mRNA accumulating during UPR, but not proteins from unprocessed *Xbp1* mRNA [107]. There is, however, little sequence homology between yeast *HAC1* and mammalian *Xbp1*, either at the nucleotide or amino acid level, except for a short stretch in the protein sequence that is rich in positively charged amino acids (arginine and lysine) corresponds to sites 29-53 in the yeast Hac1p (with sites 29-35 being the nuclear localization sequence of Hac1p). Surprisingly, both Hac1^u^p and Hac1^i^p, when expressed in mammalian cells, can induce UPR [108].

Mammalian species have evolved PERK and ATF6 pathway as two UPR branches in addition to the Ire1p/Xbp1 pathway [109,110,111]. In contrast, *Sch. pombe* has lost the Hac1p homologue and consequently the Ire1p-Hac1p pathway [89,90]. Nature has offered different options to different evolutionary lineages to respond to ER stress. A detailed evolutionary reconstruction would shed light on the loss/gain of signal transduction pathways.

### 7.2. Conservation and Diversity of Ire1p-Mediated Splicing

*HAC1* and *Xbp1* mRNA form hairpins recognized by Ire1p, and this hairpin structure is conserved among diverse evolutionary lineages from yeast to mammals [112,113] so that *HAC1* or *Xbp1* mRNA can sometimes be properly spliced by Ire1p from other species. What is particularly remarkable is that *Sch. pombe*, which no longer has a Hac1p/Xbp1 homologue, can still splice an engineered intron based on the secondary structure template shared among diverse organisms [114]. This is somewhat unexpected because Ire1p in *Sch. pombe* participates actively in RIDD [89,90] and cleaves a variety of different RNA species, with no clearly defined cleavage sites. Thus, one would expect Ire1p in *Sch. pombe* to be promiscuous in RNA substrate selection. More studies are needed to understand how *Sch. pombe* Ire1p can specifically cleave the intron at specific sites yet seem to act less discriminately against other RNA species. 

*S. cerevisiae* has only one tRNA ligase (Trl1p) which is also used to ligate cleaved *HAC1* exons. In amphioxus, three proteins (RLN, Clp1, PNK/CPDase) are involved in tRNA ligation, but only RNL can ligate cleaved *Xbp1* exons [115]. Similarly, among mammalian tRNA ligases, only RtcB is for ligating cleaved *Xbp1* exons [116,117]. *Arabidopsis thaliana* tRNA ligase (AtRlg1p) can substitute for yeast Trl1p in tRNA splicing as well as ligate the cleaved *HAC1* exons [118]. However, the resulting *HAC1* mRNA is not translated efficiently because the *HAC1* intron is circularized and likely still capable of forming the base-pair interaction with the 5’ UTR of *HAC1* mRNA to block translation initiation.

The mammalian tRNA ligase RtcB can be knocked down or even eliminated without affecting cell survival [116,117]. This raises the possibility of regulating *Xbp1* at the ligating stage. If Ire1p cleaves *Xbp1^u^*, but there is no RtcB to ligate the cleaved exon, then no mature *Xbp1^s^* would form. Is RtcB lowly expressed in non-ER-stressed cells but highly expressed under ER stress? Such a possible regulation mediated by tRNA ligase is not possible in the budding yeast. The yeast Trl1p, in contrast to mammalian RtcB, needs to be continually produced and functional for tRNA maturation, and its ligase activity consequently cannot be blocked. Another possibility that has not been examined is that cleavage sites of Ire1p in some species are degenerate but tRNA ligase picks the right exon-containing segment to ligate.

### 7.3. Diversity in Translation Control

The *HAC1* intron is conserved in various fungal species only around the splice sites that contribute to the formation of the hairpin structure recognized by Ire1p for splicing [113], but not the segment forming base-pair interaction with 5’UTR (Figure 1). This means that the mechanism of blocking translation initiation in *HAC1* is not shared beyond close relatives of the yeast. Introns in *HAC1* homologues vary dramatically in intron length. For example, *Candida parapsilosis* has an *HAC1* intron of 626 nt [84], whereas the intron is 29 nt long in *Yarrowia lipolytica* [119] and only 20 nt in *Aspergillus niger* [120]. What is particularly significant is that the short 20-nt intron in *A. niger* forms a stable stem-loop structure so it can no longer have base-pair interaction with the 5’UTR of *hacA* (homologue of *HAC1* in *A. niger*). This implies that a new translation-blocking mechanism is needed in *A. niger*, and indeed there is one through differential transcription start site. When cells are not in UPR, the transcribed *hacA* has a long 5’UTR with a GC-rich inverted repeat (18 base-pairs) that would interfere with the cap-dependent ribosome scanning to find the initiation codon. When UPR is triggered, *hacA* transcripts with a short 5’UTR missing the left half of the inverted repeat are produced. This short *hacA* mRNA can be efficiently translated to activate UPR [120]. Translation regulation mediated by different transcription start sites is also observed in filamentous fungi *A. nidulans* and *Trichoderma reesei* [121]. These two species, upon UPR induction, generate *HAC1* transcripts without a short upstream ORF. These *HAC1* messages are more efficiently translated than those with the short uORF included. However, the unconventional splicing of a 20-nt intron in these two species also contributes to increased protein production, demonstrated by different gene constructs [121].

Several Candida-related fungal species do not have an intron in their *HAC1* homologue, and would seem to require new translation-blocking mechanisms that have not yet been identified. *Sch. pombe* has Ire1p, but does not have a *HAC1/Xbp1* orthologue. However, its UPR signaling pathway does involve *IRE1* and *BIP1* which encode a major ER-chaperone [89]. The dominant hypothesis for UPR in *Sch. pombe* [89,90] is that *Sch. pombe* has recently lost its Hac1p/Xbp1 homologue, and that its Ire1p 1) participates in RIDD to reduce protein production and consequently folding load within ER, and 2) cleaves and stabilizes *BIP1* mRNA leading to increased production of Bip1p which, being a major ER-chaperone, helps to alleviate ER stress. Thus, ER chaperone proteins are increased (and UPR induced) in *S. pombe* without involving Hac1p serving as a transcription factor activating the transcription of genes encoding ER chaperones. The empirical support for a recent loss of Hac1p/Xbp1 homologue is that *Sch. pombe* Ire1p still retains the function of splicing an engineered intron with secondary structure similar to that in *Xbp1* [114]. This secondary structure is highly conserved in vertebrate *IRE1* intron [112]. However, it is also possible that another protein, very different in sequence from Hac1p, serves as a functional analogue of Hac1p in *Sch. pombe*.

Regulation of Hac1p in yeast is different from regulation of Xbp1 in metazoans. Unspliced *Hac1^u^* is hardly translated, but mammalian unspliced *Xbp1* mRNA is constitutively translated but rapidly degraded [122]. The yeast’s way of regulating the abundance of Hac1p seems more economical than the mammalian way of limiting the abundance of Xbp1.

Given the high sequence divergence, it is not surprising that yeast *HAC1* expressed in Hela and HEK293 human cell lines is generally not processed, but is translated efficiently [123]. However, other studies show that yeast *HAC1* (*yHAC1*) is correctly spliced in mammalian cells upon UPR inductions [124]. Interestingly, when *yHAC1* is expressed in mammalian cells, proteins from both spliced and unspliced *yHAC1* are active in inducing the production of chaperone proteins [108]. 

## 8. Conclusions

(Ire1p+Hac1p)-mediated UPR signaling represents a beautiful translation control mechanism created by nature. However, there are still many questions unanswered. What recognizes 3’BE of *HAC1* pre^-^mRNA and carries *HAC1* pre-mRNA to Ire1p foci to be spliced? How is translation regulated in *HAC1* homologues that do not have introns in fungal species? Is Ire1p oligomerization a consequence of Ire1p phosphorylation or the opposite? How does Ire1p dephosphorylation impact Ire1p oligomerization and *HAC1* splicing activity? How do cells exit UPR after its induction? How does this signaling system evolve and diverge in different lineages? I hope that this review will serve as an anchor for addressing all these questions in coming years.

## Figures and Tables

**Figure 1 ijms-20-02860-f001:**
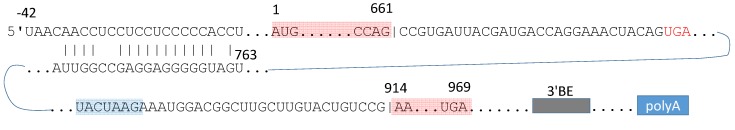
*HAC1* gene in Baker’s yeast with a 252-nt intron flanked by two exons (whose coding sequences are pink-shaded), with part of the intron forming base-pair interaction with 5’UTR to inhibit translation initiation. 3’BE is a cis-acting bipartite element signaling for *HAC1* pre-mRNA to be carried to discrete foci of Ire1p oligomers to be spliced [26]. The red ‘UGA’ within the intron stops occasional translation of the unspliced mRNA. The blue-shaded UACUAAG resembles a branchpoint site (BPS, consensus UACUAAC), but its deletion does not affect *HAC1* pre-mRNA splicing [27].

**Figure 2 ijms-20-02860-f002:**
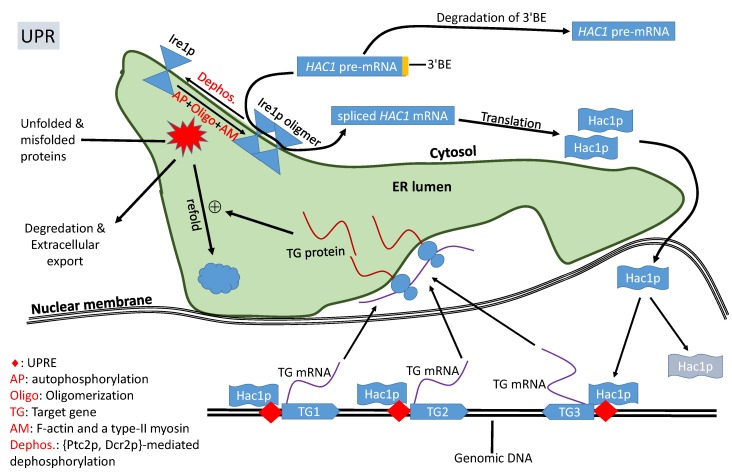
A simplified view of *IRE1+HAC1* UPR signaling pathway in yeast. Accumulation of unfolded/misfolded proteins in the endoplasmic reticulum (ER) lumen triggers the autophosphorylation and oligomerization of Ire1p which, together with Trl1p, removes the intron to release translation inhibition of *HAC1* mRNA. Hac1p enters the nucleus and transactivates target genes with a UPR element (UPRE) whose encoded proteins alleviate ER stress.

**Figure 3 ijms-20-02860-f003:**
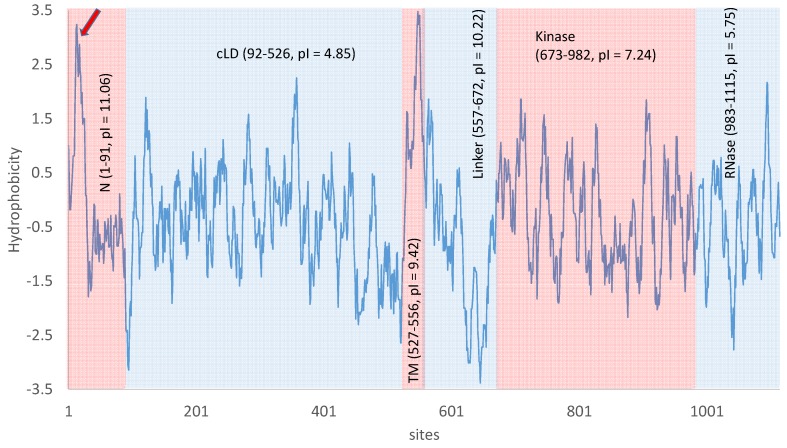
Ire1p domains, domain-specific theoretical isoelectric point (pI), and hydrophobicity [70] along the protein (sliding window of 10 amino acids). N: N-terminal domain; cLD: core luminal domain; TM: transmembrane domain; Linker: positively charged linker domain; Kinase: Kinase domain; RNase: RNase domain. The red arrow points to a highly hydrophobic stretch near the N-terminus that needs to be either buried deep into the folded Ire1p or insert into a membrane. Results generated from DAMBE [71].

**Table 1 ijms-20-02860-t001:** Contrast in splicing efficiency between some efficiently spliced yeast genes and *HAC1*, quantified by software ARSDA [59] and available in Supplemental file in Xia [58].

Gene	SystName	N_EE_	N_EI5_	N_EI3_
*RPL36B*	*YPL249C-A*	3603	10	16
*RPL18A*	*YOL120C*	2789	10	11
*RPL39*	*YJL189W*	7590	11	40
*RIM1*	*YCR028C-A*	146	0	1
*RPL43A*	*YPR043W*	4083	14	15
*RPL40A*	*YIL148W*	3928	5	19
*RPL23B*	*YER117W*	3192	19	7
*RPL2B*	*YIL018W*	2974	4	14
*RPS21B*	*YJL136C*	4192	10	14
*RPL31B*	*YLR406C*	919	4	2
*RPL25*	*YOL127W*	6536	13	22
*RPL31A*	*YDL075W*	3032	10	6
*RPL6A*	*YML073C*	3009	5	10
*RPS4B*	*YHR203C*	3411	13	6
*RPS21A*	*YKR057W*	2205	5	4
*RPS4A*	*YJR145C*	4206	8	10
*RPL33A*	*YPL143W*	4880	11	10
*SAC6*	*YDR129C*	242	1	0
*RPS7A*	*YOR096W*	5093	4	13
*HAC1*	*YFL031W*	32	844	704

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
