# Peer review of "Translation Control of HAC1 by Regulation of Splicing in Saccharomyces cerevisiae"

_ijms, 2019, doi:10.3390/ijms20122860_

Round 1
Reviewer 1 Report
Translation Control of HAC1 by Regulation of Splicing in Saccharomyces cerevisiae
Xuhua Xia
Under normal conditions, HAC1 mRNA remains translationally silent in the cytoplasm. This is because of an intervening sequence (intron) interacts with its 5’-UTR (un-translated region), thus blocking the translation at the initiation stage. Under conditions of endoplasmic reticulum (ER) stress, the ER-resident kinase/RNase Ire1 molecules become active and aggregate to form Ire1-foci. Concomitantly, HAC1 mRNA migrates towards the Ire1-foci and co-localizes with the Ire1-foci. The active Ire1 cleaves the HAC1 mRNA at two precise positions and removes the intron. Exons are then ligated by tRNA ligase to make a matured mRNA that yields an active Hac1p transcription factor. Hac1p activates plethora of genes involved in alleviating the ER stress.
In this review, author attempted to describe recent advances in the translational control of HAC1 mRNA by regulated splicing in Saccharomyces cerevisiae. Author summarizes the current views regarding the HAC1 mRNA translation and degradation. Author provides evidence that HAC1 mRNA splicing occurs mainly in the cytoplasm. Then, author deviates from the main title and describes the current view on the mechanisms of Ire1 activation and UPR signaling. There are a number of major shortcomings of this review.
1. The first line of the abstract. “HAC1 is a key transcription factor.” HAC1? Or Hac1p?
2. Only half of the review has been focused on the HAC1 mRNA translational control, thus the title does not match with the actual theme of the review.
3. Lines 80, 105. “---- a shorter protein missing the activation domain encoding in the second exon, so it does not function as a transcription factor”. This is not correct. It has been shown previously that Hac1u protein transcribed from the exon1 is also an active transcription factor.
4. Line 90. “There are three arguments in favor….” Arguments or support?
5. Figure 2. It has been shown that the target gene is translated in the ER lumen.
6. There are many sentences that need to be rephrased, including lines 226, 241, 246.
Author Response
Reviewer 1
Under normal conditions, HAC1 mRNA remains translationally silent in the cytoplasm. This is because of an intervening sequence (intron) interacts with its 5’-UTR (un-translated region), thus blocking the translation at the initiation stage. Under conditions of endoplasmic reticulum (ER) stress, the ER-resident kinase/RNase Ire1 molecules become active and aggregate to form Ire1-foci. Concomitantly, HAC1 mRNA migrates towards the Ire1-foci and co-localizes with the Ire1-foci. The active Ire1 cleaves the HAC1 mRNA at two precise positions and removes the intron. Exons are then ligated by tRNA ligase to make a matured mRNA that yields an active Hac1p transcription factor. Hac1p activates plethora of genes involved in alleviating the ER stress.
In this review, author attempted to describe recent advances in the translational control of HAC1 mRNA by regulated splicing in Saccharomyces cerevisiae. Author summarizes the current views regarding the HAC1 mRNA translation and degradation. Author provides evidence that HAC1 mRNA splicing occurs mainly in the cytoplasm. Then, author deviates from the main title and describes the current view on the mechanisms of Ire1 activation and UPR signaling. There are a number of major shortcomings of this review.
1. The first line of the abstract. “HAC1 is a key transcription factor.” HAC1? Or Hac1p?
Changed to Hac1p.
2. Only half of the review has been focused on the HAC1 mRNA translational control, thus the title does not match with the actual theme of the review.
I have thought long and hard on this, and tried to establish more links among different parts, especially in light of the other two reviewers who provided extensive comments and references on various aspects of UPR. The manuscript is now more coherent.
3. Lines 80, 105. “---- a shorter protein missing the activation domain encoding in the second exon, so it does not function as a transcription factor”. This is not correct. It has been shown previously that Hac1u protein transcribed from the exon1 is also an active transcription factor.
Thanks. I have revised it, and added the information that HAC1up is a much weaker transcription factor than HAC1ip when assessed by using β-gal as a reporter gene.
4. Line 90. “There are three arguments in favor….” Arguments or support?
Rephrased.
5. Figure 2. It has been shown that the target gene is translated in the ER lumen.
Figure redrawn.
6. There are many sentences that need to be rephrased, including lines 226, 241, 246.
I truly appreciate the feedback. The indicated sentences indeed need revision:
Line 226 has a grammatical error which has been corrected.
Line 241: the offending sentence is “The same mechanism has been documented in A. nidulans and Trichoderma reesei”. It has been changed to “Translation regulation mediated by different transcription start sites is also observed in filamentous fungi A. nidulans and Trichoderma reesei. These two species, upon UPR induction, generate HAC1 transcripts without a short upstream ORF. These HAC1 messages are more efficiently translated than those with the short uORF included.
Line 246: the offending sentence is “The functional analogue of Hac1p remains unknown.” It is in the context of the possibility that an unknown protein not homologous to Hac1p performs the same function as Hac1p. Revised.
Reviewer 2 Report
In this manuscript by Xia, the author has reviewed the translational control of HAC1 mRNA through its unconventional mRNA splicing. The author summarized recent findings on HAC1 mRNA splicing, including its translational block, mRNA targeting, the molecular mechanism of splicing, Ire1 activation/attenuation and UPR evolutionary diversification. The manuscript is clear and overall well written. However, the author missed a number of key findings made in the past few years (as detailed below). In addition, the figure 1 of the manuscript is not clearly illustrated. This manuscript will be improved by addressing the following issues.
Major comments:
1. The figure 1 illustration is difficult to digest for the readers. The ER, cytosol and nucleus are not clearly illustrated. The ER membrane, DNA and arrow lines have the same line width and color, which is difficult to distinguish. Please revise this figure.
2. In the paragraph starting from line 74, the author described the translation products of HAC1i and HAC1u. The author should include a closely related finding that the protein product of the HAC1u mRNA, but not the HAC1i mRNA, is highly unstable due to a degron at its C terminal. (Di Santo, Rachael, Soufiane Aboulhouda, and David E. Weinberg. "The fail-safe mechanism of post-transcriptional silencing of unspliced HAC1 mRNA." Elife 5 (2016): e20069.)
3. In line 111, “Inserting this 3’BE into another RNA is sufficient to direct the RNA to Ire1p foci if the RNA is not already in polysome state.” This statement is not precise. Inserting the 3’ BE AND a translation block into another mRNA is sufficient to direct the recipient mRNA into Ire1p foci.
4. In the paragraph starting from line 98, the author discussed the HAC1 pre-mRNA targeting to Ire1p foci. The author should mention that this process is also orchestrated by a positively charged region on the cytoplasmic linker domain of yeast Ire1p. (van Anken, Eelco, et al. "Specificity in endoplasmic reticulum-stress signaling in yeast entails a step-wise engagement of HAC1 mRNA to clusters of the stress sensor Ire1." Elife 3 (2014): e05031.)
5. In the paragraph starting from line 98, the author discussed the molecular mechanism of HAC1 mRNA splicing. The author should include a closely related finding on this topic: a unique mRNA structure coordinates the two splice sites and orchestrates the HAC1 mRNA splicing. In addition, by placing this mRNA structure into another mRNA, the recipient mRNA is subject to the non-conventional mRNA splicing. This finding is detailed in the following two papers: #1. Peschek, Jirka, et al. "A conformational RNA zipper promotes intron ejection during non‐conventional XBP1 mRNA splicing." EMBO reports 16.12 (2015): 1688-1698. #2. Li, Weihan, et al. "Engineering ER-stress dependent non-conventional mRNA splicing." Elife 7 (2018): e35388.
6. In the paragraph starting from line 134, the author discussed about Ire1 attenuation. However, the author didn’t include two important papers on this topic. These two papers discussed how Ire1 kinase regulates UPR attenuation. #1. Chawla, Aditi, et al. "Attenuation of yeast UPR is essential for survival and is mediated by IRE1 kinase." The Journal of cell biology 193.1 (2011): 41-50. #2. Rubio, Claudia, et al. "Homeostatic adaptation to endoplasmic reticulum stress depends on Ire1 kinase activity." The Journal of cell biology 193.1 (2011): 171-184.
7. In the paragraph starting from line 173, the author mentioned five cellular responses to restore ER homeostasis. And Ire1-mediated UPR mainly contributes to two of them. This statement is incorrect because Ire1-mediated UPR contributes to all five of them. For example, Ire1-mediated UPR transcriptionally activates ERAD in yeast (Travers, Kevin J., et al. "Functional and genomic analyses reveal an essential coordination between the unfolded protein response and ER-associated degradation." Cell 101.3 (2000): 249-258.). Ire1-mediated UPR is also involved in apoptosis in mammalian cells (Chen, Yani, and Federica Brandizzi. "IRE1: ER stress sensor and cell fate executor." Trends in cell biology 23.11 (2013): 547-555.)
8. In the paragraph starting from line 243, the author’s description on the S. pombe UPR is not precise. S. pombe has Ire1 homolog, but lacks the Hac1 homolog. S. pombe Ire1p is specialized with a more promiscuous RNase activity and alleviates ER stress through regulated-Ire1 dependent mRNA decay. This finding is detailed in two papers: #1. Guydosh, Nicholas R., et al. "Regulated Ire1-dependent mRNA decay requires no-go mRNA degradation to maintain endoplasmic reticulum homeostasis in S. pombe." Elife 6 (2017): e29216. #2 (this one already cited in the manuscript) Kimmig, Philipp, et al. "The unfolded protein response in fission yeast modulates stability of select mRNAs to maintain protein homeostasis." Elife 1 (2012): e00048.
9. In the paragraph starting from line 252, the author discussed the RNA ligase involved in the HAC1/XBP1 mRNA splicing. The author should include a recent finding, where the mammalian tRNA ligase is shown to ligate the XBP1 mRNA splicing intermediates. This is detailed in two papers: #1. Jurkin, Jennifer, et al. "The mammalian tRNA ligase complex mediates splicing of XBP1 mRNA and controls antibody secretion in plasma cells." The EMBO journal 33.24 (2014): 2922-2936. #2. Lu, Yanyan, Feng-Xia Liang, and Xiaozhong Wang. "A synthetic biology approach identifies the mammalian UPR RNA ligase RtcB." Molecular cell 55.5 (2014): 758-770.
Minor comments:
1. In abstract (line 11), it is mentioned that Ire1p form “discrete foci within ER”. The authors should be more precise as Ire1p form “discrete foci on the ER membrane”. The same suggestion also applies to line 108.
2. In line 226, “corrected spliced” should be “correctly spliced”
Author Response
-Reviewer 2
In this manuscript by Xia, the author has reviewed the translational control of HAC1 mRNA through its unconventional mRNA splicing. The author summarized recent findings on HAC1 mRNA splicing, including its translational block, mRNA targeting, the molecular mechanism of splicing, Ire1 activation/attenuation and UPR evolutionary diversification. The manuscript is clear and overall well written. However, the author missed a number of key findings made in the past few years (as detailed below). In addition, the figure 1 of the manuscript is not clearly illustrated. This manuscript will be improved by addressing the following issues.
Major comments:
1. The figure 1 illustration is difficult to digest for the readers. The ER, cytosol and nucleus are not clearly illustrated. The ER membrane, DNA and arrow lines have the same line width and color, which is difficult to distinguish. Please revise this figure.
The figure has been redrawn to eliminate ambiguity.
2. In the paragraph starting from line 74, the author described the translation products of HAC1iand HAC1u. The author should include a closely related finding that the protein product of the HAC1u mRNA, but not the HAC1i mRNA, is highly unstable due to a degron at its C terminal. (Di Santo, Rachael, Soufiane Aboulhouda, and David E. Weinberg. "The fail-safe mechanism of post-transcriptional silencing of unspliced HAC1 mRNA." Elife 5 (2016): e20069.)
I have discussed the paper in several places. Previous studies were inclined towards the hypothesis that the degradation signal is encoded in the first exon of HAC1 mRNA so both Hac1up and Hac1ip are highly unstable (half-life £ 2 min or less) because they both share the same degradation signal encoded in the first exon. In this case, there would be no need for Hac1up to have a degron of its own. The finding of Di Santo et al. raises the possibility that the degradation signal of Hac1up and Hac1ip may be independently encoded. That is, a degron in the last 10-aa of Hac1up and another degron in the last 18-aa of Hac1ip.
3. In line 111, “Inserting this 3’BE into another RNA is sufficient to direct the RNA to Ire1p foci if the RNA is not already in polysome state.” This statement is not precise. Inserting the 3’ BE AND a translation block into another mRNA is sufficient to direct the recipient mRNA into Ire1p foci.
Rephrased accordingly.
4. In the paragraph starting from line 98, the author discussed the HAC1 pre-mRNA targeting to Ire1p foci. The author should mention that this process is also orchestrated by a positively charged region on the cytoplasmic domain of yeast Ire1p. (van Anken, Eelco, et al. "Specificity in endoplasmic reticulum-stress signaling in yeast entails a step-wise engagement of HAC1 mRNA to clusters of the stress sensor Ire1." Elife 3 (2014): e05031.)
This is an excellent paper which strengthened several parts in the manuscript.
5. In the paragraph starting from line 98, the author discussed the molecular mechanism of HAC1 mRNA splicing. The author should include a closely related finding on this topic: a unique mRNA structure coordinates the two splice sites and orchestrates the HAC1 mRNA splicing. In addition, by placing this mRNA structure into another mRNA, the recipient mRNA is subject to the non-conventional mRNA splicing. This finding is detailed in the following two papers: #1. Peschek, Jirka, et al. "A conformational RNA zipper promotes intron ejection during non‐conventional XBP1 mRNA splicing." EMBO reports 16.12 (2015): 1688-1698. #2. Li, Weihan, et al. "Engineering ER-stress dependent non-conventional mRNA splicing." Elife 7 (2018): e35388.
I now included a discussion of these papers in the section on conservation and diversification of UPR.
6. In the paragraph starting from line 134, the author discussed about Ire1 attenuation. However, the author didn’t include two important papers on this topic. These two papers discussed how Ire1 kinase regulates UPR attenuation. #1. Chawla, Aditi, et al. "Attenuation of yeast UPR is essential for survival and is mediated by IRE1 kinase." The Journal of cell biology 193.1 (2011): 41-50. #2. Rubio, Claudia, et al. "Homeostatic adaptation to endoplasmic reticulum stress depends on Ire1 kinase activity." The Journal of cell biology 193.1 (2011): 171-184.
Thanks. I am particularly impressed by the experiment showing that phoshomimetic mutations of phosphorylation sites S840, S841, and T844 to Asp resulted in yeast cells failing to exit UPR after its induction. So many interesting switches in UPR......
7. In the paragraph starting from line 173, the author mentioned five cellular responses to restore ER homeostasis. And Ire1-mediated UPR mainly contributes to two of them. This statement is incorrect because Ire1-mediated UPR contributes to all five of them. For example, Ire1-mediated UPR transcriptionally activates ERAD in yeast (Travers, Kevin J., et al. "Functional and genomic analyses reveal an essential coordination between the unfolded protein response and ER-associated degradation." Cell 101.3 (2000): 249-258.). Ire1-mediated UPR is also involved in apoptosis in mammalian cells (Chen, Yani, and Federica Brandizzi. "IRE1: ER stress sensor and cell fate executor." Trends in cell biology 23.11 (2013): 547-555.)
Revised accordingly.
8. In the paragraph starting from line 243, the author’s description on the S. pombe UPR is not precise. S. pombe has Ire1 homolog, but lacks the Hac1 homolog. S. pombe Ire1p is specialized with a more promiscuous RNase activity and alleviates ER stress through regulated-Ire1 dependent mRNA decay. This finding is detailed in two papers: #1. Guydosh, Nicholas R., et al. "Regulated Ire1-dependent mRNA decay requires no-go mRNA degradation to maintain endoplasmic reticulum homeostasis in S. pombe." Elife 6 (2017): e29216. #2 (this one already cited in the manuscript) Kimmig, Philipp, et al. "The unfolded protein response in fission yeast modulates stability of select mRNAs to maintain protein homeostasis." Elife 1 (2012): e00048.
Revised accordingly.
9. In the paragraph starting from line 252, the author discussed the RNA ligase involved in the HAC1/XBP1 mRNA splicing. The author should include a recent finding, where the mammalian tRNA ligase is shown to ligate the XBP1 mRNA splicing intermediates. This is detailed in two papers: #1. Jurkin, Jennifer, et al. "The mammalian tRNA ligase complex mediates splicing of XBP1 mRNA and controls antibody secretion in plasma cells." The EMBO journal 33.24 (2014): 2922-2936. #2. Lu, Yanyan, Feng-Xia Liang, and Xiaozhong Wang. "A synthetic biology approach identifies the mammalian UPR RNA ligase RtcB." Molecular cell 55.5 (2014): 758-770.
The research in mammalian tRNA ligase RtcB is very interesting. In particular, cells survive fine with RtcB knockdown or even elimination. This raises the possibility of Xbp1 regulation by RtcB. That is, even after Ire1p cleaves Xbp1 mRNA, no mature Xbp1 mRNA is produced if RtcB does not ligate the cleaved exons. In yeast, this regulation is not possible because Trl1p needs to be continually produced and functional for tRNA maturation, and its activity therefore cannot be blocked.
Minor comments:
1. In abstract (line 11), it is mentioned that Ire1p form “discrete foci within ER”. The authors should be more precise as Ire1p form “discrete foci on the ER membrane”. The same suggestion also applies to line 108.
Changed
2. In line 226, “corrected spliced” should be “correctly spliced”
Changed
Reviewer 3 Report
Basically, this manuscript is well written and comprehensively illustrates regulatory mechanisms of the UPR mainly from the point of view of the translational control of the UPR transcription factor HAC1. However, because some important points do not seem to be described, I hesitate to recommend this manuscript to be accepted for publication in IJMS without touching on the following points.
(1) According to Di Santo et al. (2016; PMID: 27692069), the Hac1i protein is quickly digested by the ubiquitin-proteasome pathway, which avoids unwanted UPR.
(2) Attenuation of the HAC1 mRNA splicing by dephosphorylation of Ire1 upon long-term ER stress (or upon the recovery phase) is well described in Rubio et al. (2011; PMID: 21444684) and Chawla et al. (2011,; PMID: 21444691.
(3) Lines 184-186: I think that the wording in this sentence should be corrected. “Transcriptional” or “translational” control changes cellular abundance of Ire1p but does not changes the activity of Ire1p per se.
(4) As reviewed in Kimata and Kohno (2011; PMID: 21093243), I think that the most important and best-known molecular events toward activation of yeast Ire1p upon ER stress are the dissociation of Kar2p from Ire1p and direct interaction between Ire1p and unfolded proteins, which lead to oligomerization of Ire1p.
Author Response
-Reviewer 3
Basically, this manuscript is well written and comprehensively illustrates regulatory mechanisms of the UPR mainly from the point of view of the translational control of the UPR transcription factor HAC1. However, because some important points do not seem to be described, I hesitate to recommend this manuscript to be accepted for publication in IJMS without touching on the following points.
(1) According to Di Santo et al. (2016; PMID: 27692069), the Hac1i protein is quickly digested by the ubiquitin-proteasome pathway, which avoids unwanted UPR.
I have discussed the paper in several places. Previous studies were inclined towards the hypothesis that the degradation signal is encoded in the first exon of HAC1 mRNA so both Hac1up and Hac1ip are highly unstable (half-life £ 2 min or less) because they both share the same degradation signal encoded in the first exon. In this case, there would be no need for Hac1up to have a degron of its own. The finding of Di Santo et al. raises the possibility that the degradation signal of Hac1up and Hac1ip may be independently encoded. That is, a degron in the last 10-aa of Hac1up and another degron in the last 18-aa of Hac1ip.
(2) Attenuation of the HAC1 mRNA splicing by dephosphorylation of Ire1 upon long-term ER stress (or upon the recovery phase) is well described in Rubio et al. (2011; PMID: 21444684) and Chawla et al. (2011,; PMID: 21444691.
Thanks. I am particularly impressed by the experiment showing that phoshomimetic mutations of phosphorylation sites S840, S841, and T844 to Asp resulted in yeast cells failing to exit UPR after its induction. So many interesting switches in UPR......
(3) Lines 184-186: I think that the wording in this sentence should be corrected. “Transcriptional” or “translational” control changes cellular abundance of Ire1p but does not changes the activity of Ire1p per se.
I should have been more explicit. What I had in mind is that increased transcription and translation would increase total abundance of Ire1p which, by random collision with each other (without unfolded proteins) may increase the chance of accidental but unwanted Ire1p dimer formation, so Ire1 transcription and translation should not be very efficient. Increasing its transcription and translation will lead to more Ire1p and high activity of HAC1 splicing.
(4) As reviewed in Kimata and Kohno (2011; PMID: 21093243), I think that the most important and best-known molecular events toward activation of yeast Ire1p upon ER stress are the dissociation of Kar2p from Ire1p and direct interaction between Ire1p and unfolded proteins, which lead to oligomerization of Ire1p.
Thanks. That reference has strengthened the manuscript in several ways.
Round 2
Reviewer 1 Report
This review is substantially improved.
No further comment